# Polybot: Training One Policy
# Across Robots While Embracing Variability

**Jonathan Yang, Dorsa Sadigh, Chelsea Finn**
Stanford University
jyang27@cs.stanford.edu

**Abstract:** Reusing large datasets is crucial to scale vision-based robotic manipulators to everyday scenarios due to the high cost of collecting robotic datasets. However, robotic platforms possess varying control schemes, camera viewpoints, kinematic configurations, and end-effector morphologies, posing significant challenges when transferring manipulation skills from one platform to another. To tackle this problem, we propose a set of key design decisions to train a single policy for deployment on multiple robotic platforms. Our framework first aligns the observation and action spaces of our policy across embodiments via utilizing wrist cameras and a unified, but modular codebase. To bridge the remaining domain shift, we align our policy's internal representations across embodiments through contrastive learning. We evaluate our method on a dataset collected over 60 hours spanning 6 tasks and 3 robots with varying joint configurations and sizes: the WidowX 250S, the Franka Emika Panda, and the Sawyer. Our results demonstrate significant improvements in success rate and sample efficiency for our policy when using new task data collected on a different robot, validating our proposed design decisions. More details and videos can be found on our project website: https://sites.google.com/view/polybot-multirobot

**Keywords:** vision-based manipulation, multi-robot generalization

## 1 Introduction

Leveraging large datasets is essential to learning widely generalizable models in computer vision [1] and natural language processing [2]. In robotic manipulation, a promising avenue of research lies in the collection of similar extensive, in-domain datasets with the aspiration that they bestow comparable benefits. However, while past datasets have demonstrated good generalizability within the same hardware setup, applying them to different robotic configurations has proven difficult [3]. This challenge stems from four sources of variation: control scheme, camera viewpoint, kinematic configuration, and end-effector morphology. Each of these factors can vary significantly across robotic setups, leading to a large domain shift when transferring data collected on one robot platform to another. In this paper, we study the problem of how to mitigate this issue and effectively leverage robotic data across different platforms, making a stride toward learning widely-applicable policies.

In an effort to bridge the aforementioned domain gap, prior works have made advancements in enabling transfer across a subset of the factors of variation. Early works have studied cross-embodiment transfer across kinematic configurations from low-dimensional observations [4, 5]. More recently, efforts have shifted towards utilizing high-dimensional image observations, enabling transfer across robotic hardware with a fixed camera angle for 3-DoF tasks [6] and across end-effectors with a fixed embodiment [7]. Unlike these works, we do not constrain the camera viewpoint, embodiment, or low-level controller to be fixed. Instead, we propose several new design choices to align the observation and action spaces across robots, such as using wrist-mounted cameras and a shared inverse kinematics solver. Each of these choices greatly mitigate the domain shift across embodiments without compromising the generality of the robotic setup.

We integrate these design choices into a single framework that aligns the *input*, *output*, and *internal representation* spaces of our policy across embodiments. Our choice of utilizing front-mounted wrist

---

7th Conference on Robot Learning (CoRL 2023), Atlanta, USA.

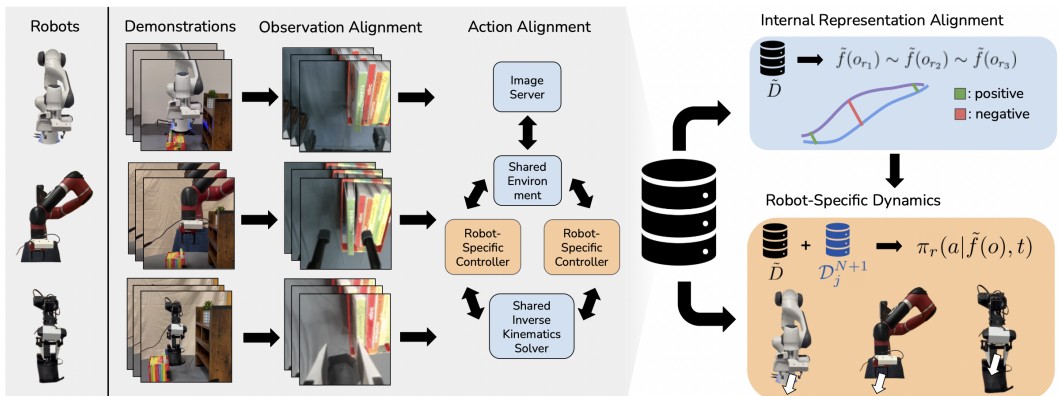

Figure 1: **Our framework for generalization across multiple robots.** We first standardize our observation space using front-mounted wrist cameras and our action space using a shared higher-level control environment. We then align our policy's internal representations using contrastive learning then finetune these representations to learn robot-specific dynamics.

cameras aligns the input (or observation) space by naturally reducing the visual variation between embodiments. This allows us to remove assumptions about collecting data only under a specific fixed angle. In order to align the output space of our policy, we employ a shared inverse kinematics solver, while allowing the lower-level controller to vary. Although we would ideally completely unify the output space by using a single abstract action space across robots, this is infeasible due to disparities in action interpretation arising from hardware discrepancies. We instead learn a multiheaded policy with robot-specific heads that capture separate dynamics. Finally, we exploit a consistent, low-dimensional proprioceptive state signal to align our policy's internal representations. This facilitates generalization across the remaining factors that may cause a domain shift.

Our main contribution is a pipeline for learning end-to-end closed-loop policies that can reuse and benefit from data collected on other robotic platforms. We empirically demonstrate that our method, **Polybot: Cross-Robot ADaptation and ALignmEnt** We first show that given a shared dataset of tasks, data from other robots for a new similar task can be transferred in a zero-shot fashion. In addition, providing as few as 5 demonstrations allows our system to achieve an average of $> 70\%$ success rate on difficult tasks which cannot be learned without data from other robots. We then show that aligning internal representations across robots can help transfer across remaining domain shifts, with our method having an average of $19\%$ higher success rate over an ablation without representation alignment. Finally, we show that our multiheaded training method achieves high success on 6-DoF tasks outperforming naive blocking controllers that achieve no success at all.

## 2 Related Work

**Learning from large datasets.** The study of utilizing diverse datasets to scale robotic learning is currently gaining momentum in the research community. Examples of large-scale real-world robot datasets include single-embodiment datasets [8, 9, 3, 10, 11, 12] and simulation benchmarks [13, 14, 15, 16]. However, these methods have typically focused on transfer with a single embodiment. To maximally reuse data from various sources, there have additionally been efforts to learn generalizable representations from sources other than robot data. One line of work is learning from human videos, including unstructured egocentric videos [17, 18, 19, 20, 21, 22, 23, 24], egocentric videos collected with a parallel gripper device [10], and in-the-wild videos [25, 26, 27]. Apart from human videos, there are works that use representations from large-scale image datasets [28, 29, 30, 31], Youtube videos [32], and multi-source data from the Internet [33]. Although these methods can improve sample-efficiency and generalizability, they are often focused on learning visual representations that can be fine-tuned for learning robot policies. Instead, we focus directly on learning visuomotor policies from robot data.

**Learning across embodiments.** A number of works have studied learning policies that transfer across embodiments. Some works in the past have focused on smaller aspects of robotic transfer,

including across 2-dimensional kinematic configurations [4], robotic models in simulation [5, 34], end-effector morphologies [7], dynamics [35, 36, 37, 38], and camera viewpoints [39]. A few works exist that focus on the task of transferring between robotic hardware [6, 40, 41]. General navigation models (GNM) demonstrates cross-embodiment transfer in navigation for wheeled vehicles, quadrupeds, and drones [40]. Hu et al. study transfer to new manipulators via a robot-aware visual foresight model trained on observations from fixed exterior camera by computing a mask [6]. In contrast to these works, we focus on transfer across a wide range of high-dimensional observations, diverse robotic control schemes, and more complex 6-DoF tasks in the low-data regime.

**Transfer learning.** Cross-robot transfer is closely related to the general area of transfer learning, where there is a large body of work on task generalization [42, 43, 44, 45, 46, 47, 48, 49, 33] and sim2real generalization [39, 50, 51, 52, 53, 54, 55]. These methods either only use a single embodiment or largely ignore cross-embodiment differences. In addition, our work is closely related to the area of domain adaptation. Previous works include using GANs [56, 57, 55, 26], domain-adversarial learning [58, 59, 60], contrastive learning [17, 21], and learning system-invariant feature representations [61, 62, 63] to transfer between two related domains. Our method focuses on the domain adaptation problem across robots by exploiting a low-dimensional proprioceptive signal.

## 3 Multi-Robot Generalization Through Domain Alignment

Our goal is to maximally reuse robotic datasets collected from one setup when deploying policies on another. Let $D_r^n = \{\{(p_0, o_0, a_0), (p_1, o_1, a_1), \ldots, (p_T, o_T, a_T)\}\}$ be a dataset containing demonstrations of robot $r$ completing task $n$. $p \in \mathcal{P}_r$ denotes the robot's proprioceptive end-effector pose, $o \in \mathcal{O}_r$ denotes the image observation, and $a \in \mathcal{A}_r$ denotes the action. We assume that there exists a shared dataset $\tilde{D} := \bigcup_{n \leq N, r \leq R} D_r^n$ of experience for $N$ different tasks and $R$ different robots. Then, given a dataset for a new task on some robotic platform $D_j^{N+1}$, $1 \leq j \leq R$, we would like to learn a policy $\pi_k^{N+1}(a|o)$, $k \neq j$ that completes the same task on robot $k$.

In order to reduce the domain gap between robots, we align the observation space, action space, and internal representations of our policy. In an ideal world, this would allow us to train a joint multitask policy $\tilde{\pi}(\tilde{a}|o, z)$ on $D_j^{N+1} \cup \tilde{D}$ where $\tilde{a} \in \tilde{\mathcal{A}}$ is a shared action with similar interpretation across robots. Prior work [40] demonstrates that training cross-embodiment policies with a unified abstract action space can facilitate learning cross-robot features by coupling similar states with similar signals. However, discrepancies in action interpretation across manipulators make this approach infeasible, so we instead train a task-conditioned multiheaded policy with $R$ heads: $\pi_r(a|\tilde{f}(o), z)$, $r \leq R$, $a \in A_r$ where $\tilde{f}$ is a shared encoder and $z$ is a one-hot task encoding. Fig. 5 and Fig. 6 in the Appendix depict this architecture.

### 3.1 Aligning the Observation Space

Ideally, an aligned observation space $\tilde{\mathcal{O}}$ would have the property that $o_1 \in D_j^n$ is similar to $o_2 \in D_k^n$ if $s_1$ is similar to $s_2$, where $s_1$ and $s_2$ denote the ground truth states of $o_1$ and $o_2$. This corresponds to $\tilde{f}(o_1) \sim \tilde{f}(o_2)$ if $s_1 \sim s_2$. To align this space as much as possible while maintaining generality of our robotic setup, we employ 3D-printed wrist camera mounts. The cameras are mounted directly in front of the robot and positioned to capture the end-effector within their field of view. However, because of variability across robots, the wrist camera mounts are not the same across robots, nor do we standardize the camera angle. Appendix A.1 shows the locations of the wrist and exterior cameras. Utilizing a wrist camera greatly simplifies the range of variation of camera positioning to two dimensions: the height of the camera with respect to the tip of the end-effector, and the angle of the camera. This is in contrast with exterior cameras, which require six dimensions to fully specify. In addition, wrist cameras provide natural robustness to distribution shifts in the environment [64]. One noticeable example of this is invariance to visual differences in robotic arms, since only the end-effector is in view as shown in the *Observation Alignment* section of Fig. 1.

### 3.2 Aligning the Action Space

To ensure a consistent action interpretation across robots, we use a shared upper-level environment. Fig. 1 depicts our control stack under the *Action Alignment* section. For an action $a_t \in \mathcal{A}_r$, we use a shared upper-level environment, responsible for processing $a_t$ and converting it into a commanded

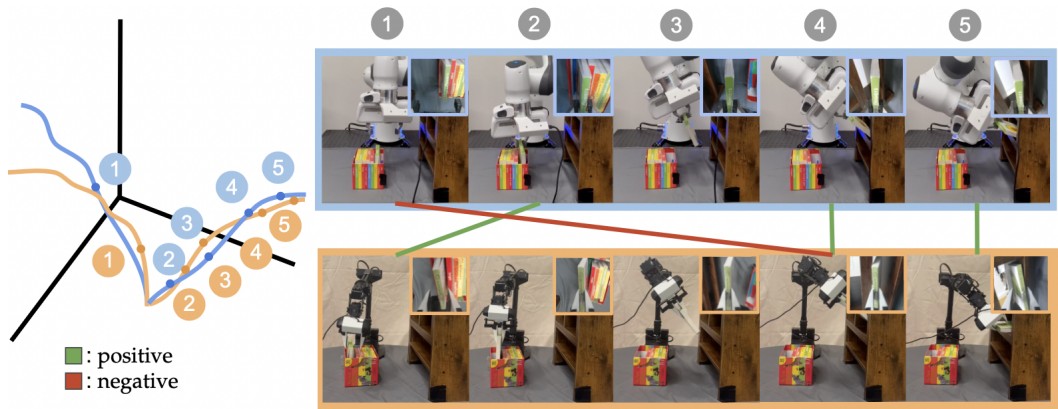

Figure 2: **Internal Representation Alignment.** This figure depicts two trajectories across different robots for the same task. Our contrastive pretraining approach maps observations with similar proprioceptive state with respect to the grasped book and cabinet together. The green lines represent example pairs of observations mapped together, while the red line represents an example pair of observations whose embeddings are pushed apart.

delta pose target $\Delta p_t^c$. Then the sum of the robot's current pose and desired target pose, $p_t + \Delta p_t^c$, is transmitted to a robot-specific controller. For modularization, each robot-specific controller $c_r$, $1 \leq r \leq R$, exposes a small, shared API to set pose targets and query end-effector poses, which is called by the upper-level environment. addition, we utilize a shared inverse kinematics solver, which processes these pose targets into joint commands. Since inverse kinematics is an underspecified problem, different solvers can cause large discrepancies in control. Our choice of a shared solver minimizes inconsistencies in interpretation of pose commands by providing a more-standardized target pose-to-joint mapping. In addition, it aligns the coordinate frame of the robots' actions, by using a consistent definition of a robot's pose with respect to its base frame. Further details of our implementations of these controllers and inverse kinematics solvers are provided in Appendix A.4.

While the above design decisions successfully aligns the poses of the robot $p_t \in \mathcal{P}_{ri}$ and ensures consistent targets $p_t + \Delta p_t^c$ are transmitted to each robot controller, it should be noted that the resulting achieved pose at the next timestep $p_{t+1}$ is not necessarily aligned. This discrepancy arises from the fact that the trajectory a end-effector follows to reach its target can differ based on the specific robot controller $c_{r_i}$ and the characteristics of the robot's hardware, even when employing a shared inverse kinematics solver. For example, consider continuous or non-blocking controllers, which interrupt the robot with a new command at regular intervals of $\Delta t$. Due to movement limitations imposed by the robot's kinematic configuration, the resulting trajectory will be highly nonlinear given the commanded pose $p_t^c$ and the current state. Consequently, even with a standardized control stack, the displacement $\Delta p_t^c$ proves insufficient for learning a shared action space with a consistent learning signal because its interpretation differs per robot.

One attempt to circumvent this issue would be to use a blocking controller, which waits for a robot to reach its target pose before issuing a new command. Firstly, we would relabel the actions in our replay buffer as the change in achieved poses $p$ and train a blocking controller to reach these poses. Since teleoperation with blocking controllers is difficult, we would instead use continuous control to collect the trajectory $\{(p_0, o_0, a_0), (p_1, o_1, a_1), \ldots, (p_T, o_T, a_T)\}$. We would then construct a new dataset $\{(p_0, o_0, \Delta p_0), (p_n, o_n, \Delta p_n), \ldots, (p_{kn}, o_{kn}, \Delta p_{kn})\}$, where $\Delta p_{kn} = p_{(k+1)n} - p_{kn}$. However, we find that even with blocking controllers, $p_t + \Delta p_t^c$ can have significant error from $p_{t+1}$. Due to limited degrees of freedom of the hardware and inaccuracies with the controller, not all $\Delta p_{t+1}$ actions are easily reachable from the current state. As a result, one robot may have significant difficulty following trajectories from other robots for tasks that require more complex 6-DoF motion. Due to these challenges, we instead use separate heads $\pi_r(a|\tilde{f}(o), t)$ to learn each robot's individual dynamics. By doing so, our policy benefits from shared visual features and overall movement directions, while enabling each head to learn the specific means to reach the desired goals.

### 3.3 Aligning Internal Representations

Aligning the internal representations of our policy can allow it to harness the advantages of training with a consistent signal, even in the absence of a unified action space. Our use of a shared kinematics solver provides a consistent proprioceptive signal $p_t$ with respect the robot's base. As a result, two (theoretical) trajectories with same exact same motion of the end effector, $T_1 \in \mathcal{D}_{r_1}^n, T_2 \in \mathcal{D}_{r_2}^n$ will differ only by some translation (i.e. for all timesteps $t$ and $p_{r_1,t}, p_{r_2,t} \in T_1, T_2$, $(p_{r_1,t} - p_{r_2,t}) = k$ for some constant $k$). Even if we do not directly act on this signal when rolling out our policy, we can still exploit it to learn better features from our observations. With this in mind, we propose to pretrain robot-agnostic features, then fine-tune to learn robot-specific controls.

In order to pretrain using the shared pose signal, we use a contrastive method that maps together similar states across trajectories and robots. We define the notion of "state similarity" by computing the changes in pose between a state and a predefined set of "fixed states" in the trajectory. Fixed states are defined as states in a trajectory that have a similar notion of task completion. For example, in quasi-static environments, these "fixed states" can be defined as states where a subtask is completed, since all successful demonstrations must contain this state. More specifically, consider a trajectory $\tau := \{(p_0, o_0, a_0), (p_1, o_1, a_1), \ldots, (p_T, o_T, a_T)\}$ and a set of fixed poses $p_f := \{p^{t_f}\}$. We define the difference between a trajectory state and fixed state by

$$d(p_i, p_{t_f})^{xyz} = p_i^{xyz} - p_{t_f}^{xyz}, \ d(p_i, p_{t_f})^{quat} = p_{t_f}^{quat}(p_i^{quat})^{-1} \tag{1}$$

where $p_i^{xyz}$ is the proprioceptive Cartesian position of the state and $p_i^{quat}$ is the orientation of the state. Since we may have more than one fixed state for trajectory, we additionally define a closest fixed state difference $d(p_i) := d(p_i, p_{t_f})$, where $t_f$ is the first timestep greater than or equal to $i$ which corresponds to a fixed states. After randomly sampling an anchor batch $A$, for each state, observation pair $(p_a, o_a) \in A$, we uniformly sample corresponding positive and negative states $(p_+, o_+) \sim U(P_{p_a}), (p_-, o_-) \sim U(N_{p_a})$ by thresholding the distances between closest fixed state differences:

$$P_{p_a} := \{(p_+, o_+); \ ||d(p_a)^{xyz} - d(p_+)^{xyz}||_2^2 < \epsilon^{xyz}, D_g(d(p_a)^{quat}, d(p_+)^{quat}) < \epsilon^{quat}\} \tag{2}$$

$$N_{p_a} := \{(p_-, o_-); \ ||d(p_a)^{xyz} - d(p_-)^{xyz}||_2^2 < \epsilon^{xyz}, D_g(d(p_a)^{quat}, d(p_-)^{quat}) < \epsilon^{quat}\} \tag{3}$$

We define $D_g(p, q)$ as $\cos^{-1}(2\langle p, q\rangle^2 - 1)$, or the geodesic distance between the two quaternions $p$ and $q$. We finally use a triplet contrastive loss to pretrain our policy: $L(o_a, o_+, o_-) = \max(0, m + ||\tilde{f}_\theta(o_a) - \tilde{f}_\theta(o_+)||_2^2 - ||\tilde{f}_\theta(o_a) - \tilde{f}_\theta(o_-)||_2^2)$ where $\tilde{f}_\theta$ is the shared encoder parameterized by $\theta$. These embeddings explicitly encourage mapping similar states together *across trajectories and robots*. This is achieved by sampling states from other rollouts in the positive buffer that correspond to similar poses with respect to a fixed pose. Fig. 2 depicts example positive and negative pair across two trajectories from different robots.

By finetuning our policy using these embeddings, we can learn robot-specific elements built upon robot-agnostic characteristics. To accomplish this, we train individualized dynamics modules $\pi_r(a|\tilde{f}(o), t)$ for each robot using a multi-headed training approach. Each head corresponds to a distinct action space that may vary across robots due to disparities in action interpretation. This is depicted in Fig. 1 by arrows denoting separate actions per robot. By adopting this technique, we circumvent potential challenges associated with trajectory matching involving blocking controllers, facilitating transferability across more intricate tasks that necessitate greater degrees of freedom.

## 4 Experiment Setup

We aim to test the following questions: (1) Does leveraging datasets from other robotic platforms enable zero-shot transfer and increase sample efficiency when learning a new task? (2) Can aligning internal representations help bridge the domain gap across robotic platforms? (3) How does our choice of multiheaded training over robot-specific dynamics compare to using a shared action space through a blocking controller? (4) How does our decision of utilizing wrist cameras compare to the past approaches of collecting data with exterior cameras? Videos of experiments can be found on our project website: https://sites.google.com/view/cradle-multirobot

**Robotic platforms.** We evaluate our method with the WidowX 250S, Franka Emika Panda, and Sawyer robot arms. All three of the robots are controlled with 6-DoF delta joint position control. For each robot, we collect 64 by 64 image observations from two sources: an exterior camera pointing at the robot, and a wrist camera mounted on the arm itself.

**Evaluation tasks.** We assume access to a shared buffer containing experience for at least one task on all three robots. Then, we collect data for a target task on two robots that we want to transfer to a third. For zero-shot evaluation, we directly train on this data and evaluate on the third robot. We also test sample-efficiency of learning a new task with few-shot experiments, where we teleoperate 5 demonstrations for the new task on the third robot.

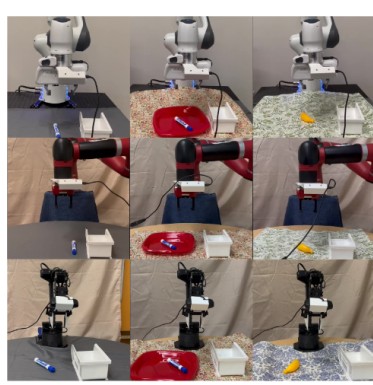

To effectively leverage data from prior tasks to solve new tasks, the prior tasks must share structural similarity with the new tasks. With this in mind, we evaluate the robot on variants for two types of tasks: standard pick/place, and shelf manipulation. In the pick/place task, the robot needs to pick up a marker and drop it into a white container from a variety of positions. In the shelf manipulation task, the robot rearranges a book from a container onto a shelf. The first set of tasks was chosen to evaluate multi-robot transfer in environments with simple dynamics and greater similarity in control from one robot to another. The second set evaluates transfer for environments that require greater degrees of motion. Each of the tasks is evaluated on 10 starting locations for the objects.

Figure 3: **Pick/Place Tasks.** The left column contains the shared pick/place task, while the other columns contain the new distractor and new object variants. For zero-shot evaluation, we include data for the shared task across all robots. For few-shot, we also include 5 demonstrations of a variant for one robot.

For each type of task, we test for both *scene generalization* and *task generalization*. We define scene generalization as generalization to a task for which there exists an mapping of observations to a previously seen task. More formally, given an optimal policy $\pi^*(a|o)$ for a previous task $T_1$, a new task $T_2$ only differs with $T_1$ by scene if there is a function $f : O \to O$ such that $\pi^*(a|f(o))$ solves $T_2$. Task generalization is more general than scene generalization, applying to new tasks which are similar to an old one, but may require different actions and controls to solve. The following are the variations that we evaluate on:

*Scenario 1 (S1): New Distractor Pick/Place*: Pick/place with a distractor object in the background
*Scenario 2 (S2): New Object Pick/Place*: Pick/place with a banana instead of a marker
*Scenario 3 (S3): New Container Pick/Place*: Pick/place putting a pen into a cup. This requires a rotation motion that is not seen in the shared dataset.
*Scenario 4 (S4): New Orientation Shelf*: Shelf Manipulation with a reversed orientation of the original container that contains the books. The motion to grasp and place the book is the same.
*Scenario 5 (S5): New Compartment Shelf*: Shelf Manipulation putting a book into a new compartment on the bottom

**Comparison methods.** For few-shot transfer, we compare our main method (denoted are *Contrastive + Multiheaded* or *Polybot* to two different baselines. The first is *Naive Multi-Robot Training*, which trains a task-conditioned multiheaded policy on exterior camera angles without contrastive pretraining. We also evaluate the *Single Robot Training* baseline, which only contains demonstrations for the target robot. We also consider two different ablations for our method: task-conditioned multiheaded training without constrastive pretraining denoted as *Ours w/o Contr.*, and constrastive pretraining with a blocking controller denoted as *Contr. + Blocking*.

**Dataset collection.** Details are provided in Appendix A.2.

## 5   Experimental Results

| Robot | Method | New Distr. (S1) | New Obj. (S2) | New Cont. (S3) | New Orient. (S4) | New Comp. (S5) |
|-------|--------|-----------------|---------------|----------------|------------------|----------------|
| | | | Pick and Place | | Shelf | |
| Franka | Polybot | **0.9** | **0.8** | **0.9** | **1.0** | **0.9** |
| | Naive Multi-Robot | 0.4 | 0.3 | 0.3 | 0.0 | 0.0 |
| | Single Robot | 0.2 | 0.2 | 0.0 | 0.0 | 0.0 |
| Sawyer | Polybot | **0.9** | **0.9** | **0.7** | **0.9** | **0.7** |
| | Naive Multi-Robot | 0.3 | 0.2 | 0.2 | 0.0 | 0.0 |
| | Single Robot | 0.2 | 0.1 | 0.0 | 0.0 | 0.0 |
| WidowX | Polybot | **0.9** | **1.0** | **0.7** | **0.8** | **0.7** |
| | Naive Multi-Robot | 0.4 | 0.1 | 0.2 | 0.0 | 0.0 |
| | Single Robot | 0.3 | 0.2 | 0.0 | 0.0 | 0.0 |

Table 1: **Few-shot multi-robot transfer results.** Given 5 demonstrations on a new task, Polybot performs significantly better than a baseline without data from other robots. In addition, our task-conditioned multiheaded policy enables the transfer of multi-robot data for shelf manipulation tasks, where a blocking controller fails.

**Utilizing data from other robots significantly improves few-shot generalization performance on a new task.** In Table 1, the success rate for Polybot demonstrates significant improvement on all tasks and all robots over single robot training. On the Franka, the results show an average of 0.56 higher success rate on *Pick/Place* (0.9, 0.8, 1.0 versus 0.4, 0.3, 0.3) and 0.95 higher success rate on *Shelf Manipulation*. (1.0, 0.9 versus 0.0, 0.0). This indicates that Polybot can effectively utilize data from other

| Robot | Method | S1 | S2 | S3 | S4 | S5 |
|-------|--------|-----|-----|-----|-----|-----|
| Franka | Polybot | **0.4** | **0.6** | 0.0 | **0.4** | 0.0 |
| | Contr. + Blocking | 0.3 | 0.3 | 0.0 | 0.0 | 0.0 |
| Sawyer | Polybot | **0.6** | **0.5** | 0.0 | **0.4** | 0.0 |
| | Contr. + Blocking | **0.6** | 0.3 | 0.0 | 0.0 | 0.0 |
| WidowX | Polybot | **0.4** | **0.4** | 0.0 | **0.5** | 0.0 |
| | Contr. + Blocking | **0.4** | 0.3 | 0.0 | 0.0 | 0.0 |

Table 2: **Zero-shot results.** Polybot can learn a new task with high structural similarity to tasks in a shared multi-robot buffer given only data from other robots.

robots to learn a new task with high sample efficiency. Notably, single-robot training fails to get any success on *Shelf Manipulation*. Qualitatively, we observe that this policy quickly falls out-of-distribution and in unable to grasp the book. For tasks with high structural similarity to those in the shared buffer such as *New Distractor Pick/Place*, training without other robot data has nonzero performance, likely due to the variation in scenes we collect in our shared dataset. However, 5 demonstrations are not enough to cover the entire distribution of object positions for a new task, leading to suboptimal performance.

**Polybot facilitates better transfer than naive task-conditioned multiheaded training on with exterior cameras.** Table 1 shows significantly higher success rates for Polybot over naive multi-robot training (0.9, 0.8, 0.9, 1.0, 1.0 versus 0.4, 0.3, 0.3, 0.0, 0.0 for the Franka). This suggests that task-conditioned multiheaded training on exterior camera struggles to utilize data from other robots, validating our hypothesis that a standardized observation space is crucial for this transfer to occur. For *Shelf Manipulation* scenarios, naive multi-robot training fails to achieve any success. For *Pick/-Place* scenarios, this method does have some success, but it is likely due to the similarities between the new task and other tasks from the same robot.

**Utilizing data from other robots allows for zero-shot generalization performance on a new task.** For tasks that have higher structural similarity to that scene in previous data for the robot such as scene generalization tasks described in the experiment setup section, we see good success rate for zero-shot multi-robot transfer. Table 2 showcases this, with both Polybot having success rates of 0.4, 0.6, 0.4 and the blocking controller having success rates of 0.3, 0.6, 0.4 for the *New Distractor Pick/Place task*. Without data from other robots, the success rate would be near 0 because the new task would not be present in the replay buffer. For Scenario 4, or *New Orientation Shelf*, Polybot has nonzero performance while the blocking method doesn't. This is because while both methods are able to pick up the book from the reversed bookshelf, the blocking controller struggles with placing the book onto the shelf.

**Aligning the internal representations of the observations between robots leads to learning more generalizable features.** Table 3 compares the performances of Polybot with an ablation of the

| Robot | Method | S1 | S2 | S3 | S4 | S5 |
|-------|--------|----|----|----|----|----|
| Franka | Polybot | **0.9** | **0.8** | **0.9** | **1.0** | **0.9** |
| | Polybot w/o Contr. | 0.8 | 0.5 | 0.7 | 0.6 | 0.7 |
| Sawyer | Polybot | **0.9** | **0.9** | **0.7** | **0.9** | **0.7** |
| | Polybot w/o Contr. | 0.7 | 0.5 | 0.4 | 0.6 | 0.7 |
| WidowX | Polybot | **0.9** | **1.0** | **0.7** | **0.8** | **0.7** |
| | Polybot w/o Contr. | 0.8 | 0.7 | **0.7** | 0.6 | 0.7 |

| Method | S1 | S2 | S3 | S4 | S5 |
|--------|----|----|----|----|----|
| Polybot | **0.9** | **0.8** | **0.9** | **1.0** | **0.9** |
| Contr. + Blocking | 0.8 | 0.6 | 0.0 | 0.0 | 0.0 |
| Polybot | **0.9** | **0.9** | **0.7** | **0.9** | **0.7** |
| Contr. + Blocking | **0.9** | **0.9** | 0.0 | 0.0 | 0.0 |
| Polybot | **0.9** | **1.0** | **0.7** | **0.8** | **0.7** |
| Contr. + Blocking | 0.8 | **0.9** | 0.0 | 0.0 | 0.0 |

Table 3: **Ablation: ours vs ours without contrastive.** Our contrastive pretraining and multiheaded finetuning approach provides an average of $19\%o$ improvement on few-shot transfer for a new task over regular multiheaded training.

Table 4: **Ablation: Ours vs contrastive + blocking** Although a blocking controller has similar few-shot performance to Polybot on simple Pick/Place variants, it struggles with tasks that require 6-DoF motion.

contrastive pretraining phase. Aligning the internal representation of the policy causes an average of $19\%$ increase in performance over a baseline without contrastive pretraining. This suggests that aligning the internal representation of the policy assists in learning features that generalize across robots. We hypothesize that training on an aligned proprioceptive signal across robots. Notably, multiheaded training alone seems to have reasonable performance on all tasks, albiet lower than our method.

**Multiheaded policies can better learn from 6-DoF, cross-robot demonstrations over blocking controllers.** Table 9 shows that on the *Shelf Manipulation tasks*, Polybot achieves high success rates $1.0, 0.9$ on the Franka compared to zero success rate on the blocking controller. In addition, in the *New Container Pick/Place*, Polybot has a $0.9$ success rate over the $0.0$ with blocking. This large discrepancy is explained by the inability of one robot to precisely imitate another robot. For instance, variations in the length of the wrist link can lead to large discrepancies in the radius of the rotation necessary to perform the *Shelf Manipulation* task. To verify this, we provide the error over timestep of imitating each task in the Appendix. Notably, the blocking controller has high performance on *S1* and *S2* due to lower discrepancy between robots for translational movement.

## 6 Discussion

**Summary.** We have developed a method, Polybot, that efficiently learns new tasks using data collected on other robots. This is enabled through careful design decisions driven by a fundamental observation: transferring data across domains requires aligning the domains as much as possible without making assumptions that limit their applicability. For example, Polybot uses wrist cameras, which can be mounted on a wide range of robots, while exhibiting significantly less variation than exterior cameras, even if the mounting position is not fixed. In addition, Polybot uses a shared higher-level action representation and varying lower-level controller, which can align the policy's actions while accommodating the diversity of robotic setups. Finally, our choice of contrastive loss can align internal representations across robots. Despite the simplicity of each design decision, their combination sufficiently aligns our multi-embodiment dataset to enable cross-embodiment transfer. Polybot achieves over a $70\%$ success rate on all tasks, outperforming naive multi-robot training.

**Limitations and Future Work.** One limitation of our approach is that our method requires a shared dataset between robots to learn their correspondences. As a result, our method is not able to transfer policies to a new robot with no demonstrations. Our method also does not allow for zero-shot transfer on tasks with different motion than seen in the shared dataset. In addition, the scope of our evaluation has been on parallel-jaw robotic manipulators. Generalization can become more difficult with a more diverse set of end-effectors, although our method does not preclude this type of transfer. Our use of egocentric cameras may lead to difficulties in partially observable settings, such as settings without a clear view of the end-effector. Finally, our representation alignment relies on a scalable method to compute fixed states across all trajectories, which may not apply for all manipulation tasks. In the future, we plan to scale our datasets, as we believe that can allow for more new tasks, camera-viewpoints, and end-effectors to be in-distribution.

**Acknowledgments**

We thank Tony Zhao, Moojin Kim, and Alexander Khazatsky for the numerous discussions about real-world robot learning and Kyle Hsu, Hengyuan Yu, and Suvir Mirchandani for their helpful feedback. This research was supported by the Office of Naval Research grants N00014-22-1-2621 and N00014-22-1-2293.

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

# A    Appendix

Further details and videos of experiments can be found on our project website: https://sites.google.com/view/polybot-multirobot

## A.1    Robotic Setup

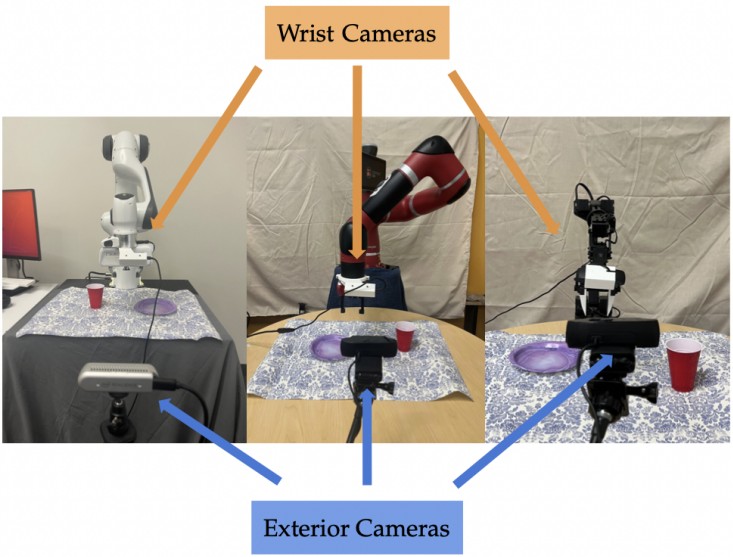

Figure 4: **Our robotic setups.** For each robot, we collect data with both a wrist camera and exterior camera. The cameras are Logitech C920s and Zeds. Although these cameras do have slight differences in brightness and contrast, this does not seem to affect results.

## A.2    Dataset Collection

We collect two types of datasets: a shared dataset containing data from similar tasks for all three robots and a target dataset containing a new task from one robot platform which we want to transfer to other platforms. For the purposes of evaluation, our shared dataset consists of the original tasks we defined above. For each variant, we collect data on 3 diverse scenes and backgrounds to ensure that the resulting policies have some degree of robustness to changes in the environment. In order to provide diversity to visual observations, we use cups, plates, and wallpapers with intricate patterns in the background. By collecting our datasets for each task over 3 variations of this scene, we ensure that our policy is robust to changes in lighting conditions. Overall, our dataset contains 6 tasks over 3 robots with 3 different backgrounds per task and 50 demonstrations per (scene, robot, background) combination collected over the course of 60 hours.

## A.3    Higher-Level Environment Details

Our higher-level environment has of a shared image server and action processor between robots. We use delta position control as our action space. This is parameterized by a 7-dimensional vector consisting of 3 translational dimensions, 3 rotational dimensions, and 1 dimension indicate the percentage to close the parallel end-effector. The code for processing an action before sending it to the lower-level controller is shown below:

```
def step(self, action):
    start_time = time.time()

    # Process Action
    assert len(action) == (self.DoF + 1)
    assert (action.max() <= 1) and (action.min() >= -1)
```

```
pos_action, angle_action, gripper = self._format_action(action)
lin_vel, rot_vel = self._limit_velocity(pos_action, angle_action)
desired_pos = self._curr_pos + lin_vel
desired_angle = add_angles(rot_vel, self._curr_angle)

self._update_robot(desired_pos, desired_angle, gripper)

comp_time = time.time() - start_time
sleep_left = max(0, (1 / self.hz) - comp_time)
time.sleep(sleep_left)
```

Given a delta position, angle, and gripper command, our environment first normalized and clips the commands to ensure that large actions are not sent to the robot. Then, we add the delta position to our current pose and the delta angle to our current angle. We pass the position and angle into our lower-level robot controller.

## A.4 Robot-Specific Controller Details

Each robot-specific controller provides the following API to the higher-level environment:

```
def update_pose(pos, angle):

def update_joints(joints):

def update_gripper(close_percentage):

def get_joint_positions():

def get_joint_velocities():

def get_gripper_state():

def get_ee_pose():
```

The functions update_pose, update_joints and update_gripper set targets for moving the robot.
For each lower-level controller, we use a shared inverse kinematics solver to take the target poses in update_pose and convert them into joint targets. For simplicity, we use a Pybullet-based solver and URDF model specifications for each of our robots to compute target joint positions from Cartesian poses. When computing inverse kinematics through Pybullet, we manually set the joint limits of each robot, since the solver does not automatically consider these limits by itself. We also use the IK solver to give us joint positions, joint velocities, gripper states, and end-effector poses. This allows us to use a standarized coordinate from with respect to the robot's base to get a robot's Cartesian coordinates.

For each robot, we implement two controllers: a blocking version and a nonblocking version. The blocking controller waits for an entire movement command to finish before executing the next command. Meanwhile, the nonblocking or continuous controller continuously interrupts the robot with a new target pose every fixed period of time.

## A.5 Network Architecture

## A.6 Contrastive Learning Details

We train our encoder with a triplet loss of margin $m = 0.5$.

$$L(o_a, o_+, o_-) = \max(0, m + ||\tilde{f}_\theta(o_a) - \tilde{f}_\theta(o_+)||_2^2 - ||\tilde{f}_\theta(o_a) - \tilde{f}_\theta(o_-)||_2^2)$$

We provide nearest neighbor lookup for our robot below. We first embed the left image via our encoder. Then, we embed all observations in a dataset for a different robot. For example, in the top-left image, we use the shelf manipulation dataset with only Franka data. Then, we compute the

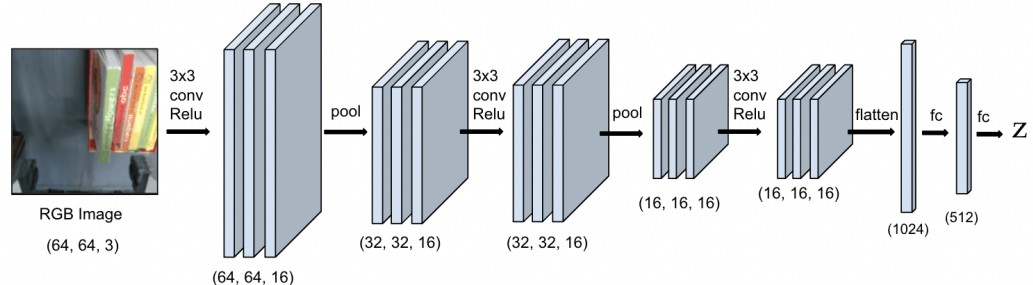

Figure 5: **Our encoder architecture.** We parameterize our encoder as a CNN. The convolutional layers are flattened and then fed into two MLP layers to get a representation $z$. In order to learn correspondence between robots, we train this encoder with a contrastive loss. We use random crop and color jitter as image augmentations for our encoder.

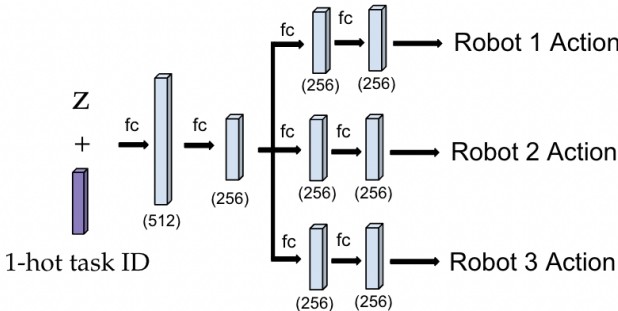

Figure 6: **Our decoder architecture.** The output of our encoder z is concatenated with a one-hot task index and fed into the decoder. This task index specifies a task either in the shared buffer, or a new task which we want to achieve. After passing the input through two MLP layers, we feed in into three-robot specific heads for each of the robots we are evaluating on.

| Attribute | Value |
|---|---|
| Input Width | 64 |
| Input Height | 64 |
| Input Channels | 3 |
| Kernel Sizes | [3, 3, 3] |
| Number of Channels | [16, 16, 16] |
| Strides | [1, 1, 1] |
| Paddings | [1, 1, 1] |
| Pool Type | Max 2D |
| Pool Sizes | [2, 2, 1] |
| Pool Strides | [2, 2, 1] |
| Pool Paddings | [0, 0, 0] |
| Image Augmentation | Random Crops/Color Jitter |
| Image Augmentation Padding | 4 |

Table 5: **CNN hyperparameters for our policy encoder.** Our CNN uses 64 by 64 images, which passes through through 3 convolutional layers. Each layer has a $3by3$ kernel with 16 channels. We augment our architecture with random crop and color jitter.

embedding with the closest $l2$ distance from the embedding of the left image. Note that our method also aligns trajectories with same robot.

| Hyperparameter | Value | Hyperparameter | Value |
|---|---|---|---|
| Batch Size | 64 | Batch Size | 64 |
| Number of Gradient Updates Per Epoch | 1000 | Number of Gradient Updates Per Epoch | 1000 |
| Learning Rate | 3E-4 | Learning Rate | 1E-4 |
| Optimizer | Adam | Optimizer | Adam |

Table 6: **Hyperparameters.** The left table contains hyperparameters for behavior cloning, and the right table contains hyperparameters for contrastive learning.

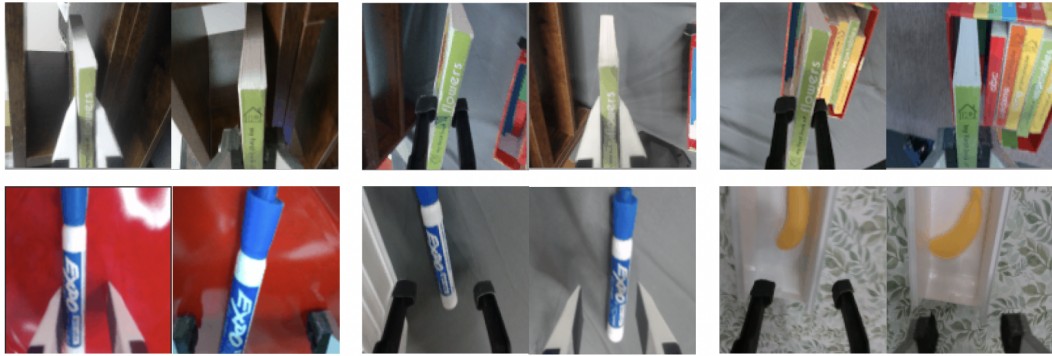

Figure 7: **Contrastive Nearest Neighbors.** This figure shows nearest neighbors examples across the three robots for embeddings from our pretrained encoder. These examples are computed for both shelf and pick/place trajectories.

## A.7 Shelf Tasks

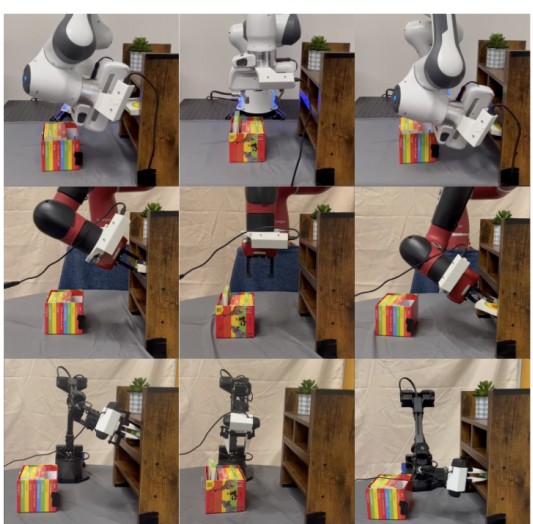

Figure 8: **Shelf Tasks.** The original shelf tasks consists of placing the book on the top compartment. The first target task requires doing the same from a reversed book container, while the second tasks requires placing the book in the lower compartment. The tasks in the first column are part of the shared dataset while the second and third are target tasks to test transfer.

## A.8 Error between Commanded Delta Pose Target and Achieved Delta Pose

The following figures depict a plot of the $l2$ norm between the translational components of the delta commanded pose targets and achieved delta poses for demonstration trajectories across 3 robots. At each timestep, the environment receives a delta commanded pose target, which gets added to the robot's current pose then sent to the lower-level controller. Although the controller defines a trajectory to reach this target pose, due to errors in the inverse kinematics solver and limitations on movement imposed by the hardware, it may not reach the pose. We plot the error for each timestep across a trajectory from a Pick/Place task and one from a Shelf Manipulation task. Expectedly, the WidowX has the highest average error, followed by the Sawyer then the Franka. This error varies wildly between robots and timesteps, causing the commanded delta pose to be highly unpredictable from the achieved delta pose.

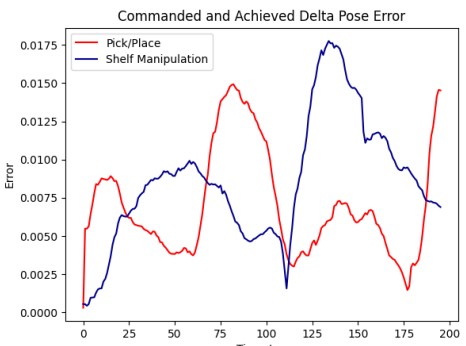

Figure 9: **Action interpretation error for the Franka.**

Figure 10: **Action interpretation error for the Sawyer**

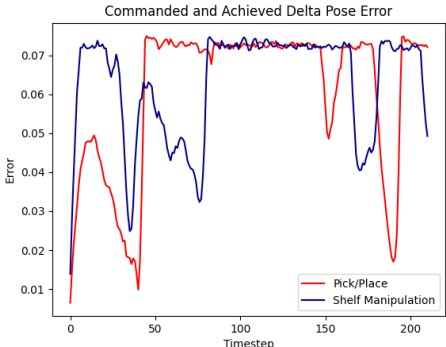

Figure 11: **Action interpretation error for the WidowX.**

### A.9 Ablation: Wrist Camera Variation

In order to more comprehensively evaluate the effect of wrist-camera variation, we have 3D-printed 2 new wrist-camera mounts for the WidowX250S. We have then collected data for the shared **Pick-/Place** task as well as **New Distractor Pick/Place** task variant. Here are the results

**Viewpoint 1 (V1): Original Mount**: The original camera mount used on the WidowX 250S for our experiments. The mount is 20 degrees from vertical.
**Viewpoint 2 (V2): Original Mount + Masked Gripper:** The original camera mount used on the WidowX 250S experiments with the bottom part of the image masked out.
**Viewpoint 3 (V3):** 35 **Degree Mount** A wrist-camera viewpoint that is 35 degres from the vertical.
**Viewpoint 4 (V4):** 50 **Degree Mount** A wrist-camera viewpoint that is 50 degres from the vertical.
**Viewpoint 5 (V5):** 50 **Degree Mount + Change in Height**: The 50-degree wrist-camera mount is placed lower on the WidowX.

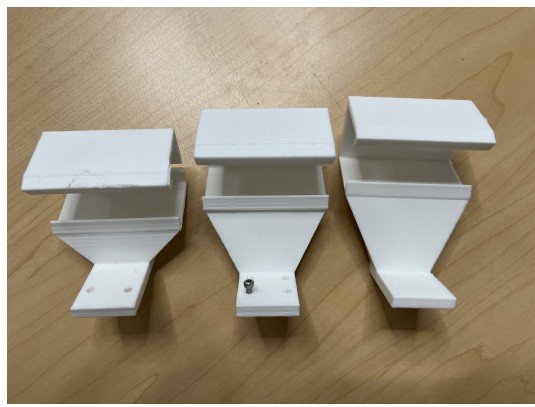

Figure 12: **Our wrist camera mounts.** The follow depicts the original, 35-degree and 50-degree mounts.

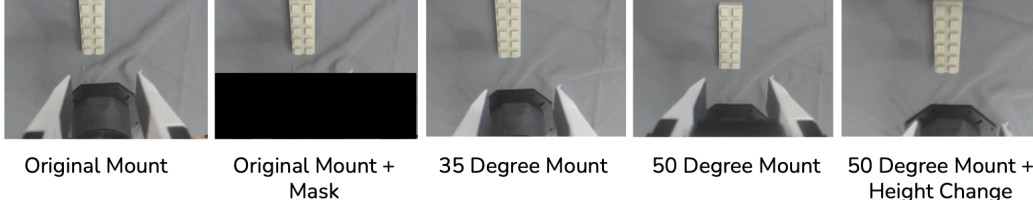

Figure 13: **Our wrist camera viewpoints.** This figure depicts the viewpoints we used for our ablations. The lego block is in the same place directly under the gripper for all mounts. Different wrist camera angles can cause the same object to appear in different parts of an egocentric image.

| Method | V1 | V2 | V3 | V4 | V5 |
|--------|----|----|----|----|----|
| CRADLE | 0.8 | 0.8 | 0.9 | 0.8 | 0.7 |

Table 7: **Ablation: Wrist Camera Viewpoint** The table depicts the few-shot performances of the new-distract Pick/Place task with new distractor viewpoints. Each variation was evaluated 10 times on a wide variety of angles.

Our results show consistently high success rate on a target task for the variation in viewpoint we evaluated on. Notably, even though the location of the object on the image changes, CRADLE is still able to learn good correspondences and transfer experience to these camera angles. We expect

that by training on a larger shared dataset with more variations in camera angle, we will be able to finetune directly on new camera viewpoints without requiring shared data.

Interestingly, the policy performs well on the Pick/Place even with the bottom part of the image cropped out. In order to make these experiments work, we had to include the robot's proprioceptive information alongside the 1-hot task ID and latent variable to the decoder. This allows the state to maintain full observability on the Pick/Place tasks. With more than 1 viewpoint that does not contain the gripper, the policy may have problems determining the camera angle from the image alone. This is because there one can achieve the same image by either changing the camera angle or moving the robot's end-effector.

### A.10 Ablation: End-Effector Size Variation

In order to consider the effect of variations in end-effector size, we have ran experiments on Pick-/Place tasks with a larger and smaller gripper. Although we reuse data with the larger gripper, we collect new data on both the shared Pick/Place task as well a small number of demonstrations for each task variant using the smaller gripper. Similar to our main experiments, we finetune our policy with 5 demonstrations for task variant. In order to succeed, a robot has to slightly adapt its actions in order to transfer robotic data collected from the larger gripper to the smaller. If incorrect features are transferred across these two settings, the policy may attempt to grasp the object too early.

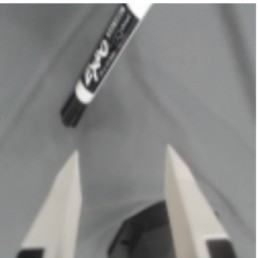 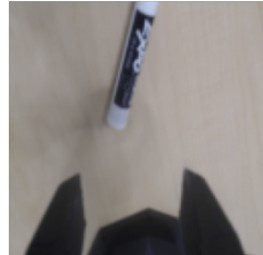

Larger Gripper           Smaller Gripper

Figure 14: **Egocentric Viewpoints for the Larger/Smaller Grippers.** The fingers on the left are a larger 3D-printed variations of the smaller ones designed by Trossen Robotics on the right.

| Method | S1 | S2 | S3 |
|--------|-----|-----|-----|
| CRADLE | 0.9 | 1.0 | 0.7 |

Table 8: **Ablation: End-Effector Size Variation**: CRADLE achieved an average of $87\%$ success on few-shot generalization to a new gripper.

Our experiments show that CRADLE has high success rate on Pick/Place tasks with both small and large grippers. These results are in-line with the results we see from transfer across the Franka's Sawyer's and WidowX's end effectors. Since the grippers are different sizes and widths, the policy will need to learn how to adapt to their individual constraints in order to get effective transfer performance.

### A.11 Ablation: Joint Egocentric and Exterior Training

Although our settings do not require partial observability, to provide evidence that CRADLE is able to transfer information with joint egocentric and exterior training, we have runs New Distractor Pick-/Place experiments with both joint and exterior camera training. In order to process this information, we stack these viewpoints together channel-wise before passing it through the convolutional neural network encoder. The following table describes the results:

| Method | S1 | S2 | S3 |
|--------|-----|-----|-----|
| CRADLE | 0.7 | 0.7 | 0.6 |

Table 9: **Ablation: Joint Egocentric and Exterior Training** The table depicts the performances of the new-distract Pick/Place task with new distractor viewpoints. Each variation was evaluated 10 times on a wide variety of angles.

Although CRADLE is able to achieves $> 60$ success rate with both egocentric and exterior camera, we see that its performance suffers. We believe that this is due to difficulties aligning the third-person observations. Although wrist-camera perspectives look similar across embodiments, third-person perspectives can have vary based on the a robot's appearance. We believe that by pretraining with more

