# OpenReview forum: "Polybot: Training One Policy Across Robots While Embracing Variability"
_robot-learning.org/CoRL/2023/Conference — CoRL 2023 Poster_

### Official Review · Reviewer_Fa2w · 2023-07-10

**Confidence:** 4
**Originality:** Fair
**Technical Quality:** Good
**Clarity Of Presentation:** Fair
**Impact:** 3

**Recommendation:**

Weak Accept: I recommend accepting the paper, but will not argue for my recommendation if the majority of other reviewers have a different opinion.

**Review:**

++: Strength, --: Weakness

**Novelty & Significance**

++ Cross-embodiment imitation learning is a hard problem that is still an open question. I like the idea of CRADLE such that different engineering tricks can realize alignments in observation, action, and state representation.

-- However, I think that this idea has its own downside. Essentially, CRADLE bypasses the correspondence problem between different embodiments by representing the state on task space, egocentric viewpoint, and custom controller that requires known robot models. I believe this limits the framework to object-centric manipulations having chain-like robot arms with a single end-effector. Scaling up to human - robot imitation or single-arm robot - bi-arm robot imitation is impractical.

-- Furthermore, embodiment-specific mapping is not addressed. For example, consider the imitation learning task of navigating through a hole, robot demonstration moving through a narrow hole will be hard to transfer with CRADLE’s egocentric end-effector view.

-- Another limitation is in the contrastive learning method in this setting. To sample the positive and negative samples for each anchor point, CRADLE has to compute the distance from the anchor points to all state samples in the dataset, which hinders the scaling of this framework to a large dataset.

Overall, I think that there should be a discussion in the paper on the egocentric characteristics with the above points. It is important for the imitation learning community for this paper to have a clear scope. Other limitations are already mentioned by the authors.

**Quality & Clarity**

I highlight some comments on the paper presentation.

++ The paper is easy to follow and has a clear structure.

++ Fig. 1 and 2 assist the paper’s understanding well.

++ I believe this paper has already good experiment setups. However, more experiments on new tasks such as pouring, cleaning, etc. beyond pick&place task is appreciated.

-- Further explanation in Section 3.2 is needed. What is the blocking controller? How is the shared inverse kinematic solver designed?

-- There is a typo on the equation at line 178.

-- There is an incomplete discussion on line 268.

-- There are unnumbered equations.

**Quality Of The Limitations Section:**

Additional details required

**Questions For Rebuttal:**

1. Discussion on the possible further applications of CRADLE to different cross-embodiment imitation learning settings.
2. How to efficiently implement the positive and negative sampling for contrastive learning in this setting?
3. Further elaborations for Section 3.2 is desirable.

**Robotics Focus:**

Sufficient demonstration on hardware

**Summary Of Paper:**

This paper proposes CRADLE - a framework for cross-embodiment adaptation and alignment. CRADLE is an imitation learning framework that tries to learn a skill of a chosen robot from different robot demonstrations. To align the observation space, the camera is mounted at the end-effector of each robot. It collects a dataset of demonstrations for a set of tasks from a set of robots, then trains a neural network policy with $R$ heads corresponding to $R$ robots to accommodate their different dynamics with shared visual presentation. During the training phase, to align the state latent representation, CRADLE adopts the contrastive learning method with the state distance defined on task space. The method’s efficacy is shown in real-world pick&place tasks between multiple robots.

**Summary Of Recommendation:**

Overall, due to:
- incomplete paper discussions
- computation drawback that hinders this method's significance
- lack of clarity in some sections

I recommend rejection but I can change my assessment based on the authors' rebuttal.

------ Post Rebuttal ------

The authors addressed all my concerns. I decided to raise the score to Weak Accept.

---

### Official Review · Reviewer_D1Nx · 2023-07-19

**Confidence:** 4
**Originality:** Good
**Technical Quality:** Good
**Clarity Of Presentation:** Fair
**Impact:** 3

**Recommendation:**

Weak Accept: I recommend accepting the paper, but will not argue for my recommendation if the majority of other reviewers have a different opinion.

**Review:**


Strengths

1) This paper considers an important direction, which will draw the attention of the research community.

2) Experiments show impressive results of the proposed method.


Weaknesses:

1) Some technical details need to be clarified.

    The reviewer is confused about why the observations are the same since the height and angle of the cameras are different. Did the authors set a fixed height and angle, so that adjust all observations based on them?

    The reviewer is confused about the problem setting since the notations are confusing.

      a) In Line 102, 'would like to learn a policy $\pi_{k,N+2}(a|o)$', why not using $\pi_{j,N+1}(a|o)$? Does this mean the goal is to learn a new policy on a new task N+2 using robot k, given a dataset on the other new task N+1 using robot j?

      b) In Line 110, does $t$ denote the time step? Then why $t$ is included in a policy? However, in Line 102, $t$ is not included in a policy denotation.

    The action alignment part is also confusing. It seems like the action is converted to the state (observation) change, but under what space? The aligned observation space? Also, the reviewer is not familiar with inverse kinematics solver, it would be better to describe how and why ' a shared inverse kinematics solver could minimize inconsistencies in interpretation of pose commands'.

2) Missing some discussions of related works.

3) More limitations should be provided in the paper.

Questions,

The reviewer is curious about whether the proposed method is applicable to more diverse settings. For example, the two control tasks are the same, i.e., pushing a box to the target location. However, the friction may be different. In one task the box is on smooth ground, while in the other one, the box is on a carpet. In such a case, how to adapt the policy to the new setting? From the current version, the action alignment does not consider this.

Minors,

Some typos need to be fixed, for example, in Line 139, 'a end-effector follows...' -> 'an end-effector follows...'


Some related works are not investigated in this paper, for example [1-4].

[1] Learning cross-domain correspondence for control with dynamics cycle-consistency.

[2] Reinforcement Learning with Videos: Combining Offline Observations with Interaction.

[3] Cross-domain Adaptive Transfer Reinforcement Learning Based on State-Action Correspondence.

[4] Sim-to-real robot learning from pixels with progressive nets.


**Quality Of The Limitations Section:**

Additional details required

**Questions For Rebuttal:**

Please refer to the pros and cons part.

Questions,

The reviewer is curious about whether the proposed method is applicable to more diverse settings. For example, the two control tasks are the same, i.e., pushing a box to the target location. However, the friction may be different. In one task the box is on smooth ground, while in the other one, the box is on a carpet. In such a case, how to adapt the policy to the new setting? From the current version, the action alignment does not consider this.



**Robotics Focus:**

Sufficient demonstration on hardware

**Summary Of Paper:**

This paper proposes a new method to transfer the trained policy and also reuse the datasets across different robots. The visual observation alignment and action alignment are proposed, in addition, to solve the domain shift, the internal representation of the policy is also learned via contrastive learning.


**Summary Of Recommendation:**

A weak rejection is given based on the current status. But the reviewer is willing to increase the score if all concerns are resolved during the rebuttal.

My concerns have been addressed after reading the rebuttals, thus increasing my score to weak accept.

---

> ### Author Response · Authors · 2023-08-15
> **Follow Up**
>
> Please let us know if our response has addressed your concerns, or if if there's anything we can further clarify. We would be happy to answer any questions in more detail. Thank you!

---

### Official Review · Reviewer_tpTQ · 2023-07-20

**Confidence:** 4
**Originality:** Very Good
**Technical Quality:** Very Good
**Clarity Of Presentation:** Excellent
**Impact:** 4

**Recommendation:**

Weak Accept: I recommend accepting the paper, but will not argue for my recommendation if the majority of other reviewers have a different opinion.

**Review:**

**Strengths**
- The paper considers several practical challenges and systematically addresses them one by one.
- The paper is well written and easy to follow.
- The experimental results are presented and discussed in a comprehensive manner.
- The results conclusively demonstrate both the effectiveness of the approach and the need for each of its components.

**Weaknesses**
- The tasks and robots used in the experiments are not that different. Indeed, as the authors acknowledge, the method does not solve the correspondence problem completely. Instead solves it within the context of a set of similar tasks on .
- The method seems to be restricted to datasets collected with the wrist mount camera. It is unclear if results would be affected even if a different wrist-mounted camera is used.
- Minor: 1. There might be a mismatch between the colors in Fig.2 and the caption. 2. Both "i" and "n" are used to denote task index first paragraph on Section 3

**Quality Of The Limitations Section:**

Limitations are addressed clearly

**Questions For Rebuttal:**

Please address my comments from the "Weaknesses and Suggestions" section of my review comments and let me know if there are any substantive misunderstandings on my part. In particular, please comment on how this method would be able to handle end-effectors that look significantly different, like multi-fingered hands or suction cups (ignoring, of course, the challenges of low-level control).

**Robotics Focus:**

Sufficient demonstration on hardware

**Summary Of Paper:**

The paper proposes a framework to enable learning a morphology-agnostic policy to facilitate learning from datasets accumulated across different robots. It does this by aligning demonstrations from different robots using 3 steps: i) align the visual input by using a wrist-mounted cameras, ii) align action space by using multi-head policies to capture robot-specific dynamics, and iii) align internal representations using contrastive learning.

**Summary Of Recommendation:**

The paper is well written and easy to follow. It addresses an important and timely problem: being able to leverage datasets from other robots. The proposed method is simple and effective, and the experimental baselines are well-designed. However, the experiments are conducted on a fairly limited set of tasks that are very similar to one another.

---

### Official Review · Reviewer_LHXF · 2023-07-23

**Confidence:** 5
**Originality:** Good
**Technical Quality:** Very Good
**Clarity Of Presentation:** Very Good
**Impact:** 4

**Recommendation:**

Weak Accept: I recommend accepting the paper, but will not argue for my recommendation if the majority of other reviewers have a different opinion.

**Review:**

This paper delves into a critical issue in robot learning: utilizing data across various robotic morphologies for efficient training related to manipulation tasks. The reviewer acknowledges the considerable complexities and challenges involved in this project and commends the authors' determination to address them. The paper's strengths are manifold.
- Firstly, it introduces a robust method to synchronize the observation and action spaces among different robotic entities. While employing cartesian control has proven to be more invariant to kinematics than joint control, incorporating this concept into a learning framework is an outstanding engineering accomplishment.
- Secondly, the paper suggests a contrastive learning strategy to enhance the shared visual representation learning among various robots. This approach is quite intuitive given the design of robot-specific dynamics. The perceptual task involves understanding the spatial relationship between the object and the gripper, which can be common across morphologies, while action prediction is dependent on the individual controller characteristics of each hardware piece.

However, the paper could benefit from enhancements in a couple of areas:
1. The alignment method in contrastive learning could be improved to better define positive and negative examples. The current method is heavily reliant on the end-effector pose, a strong assumption for manipulation tasks. More consideration should be given to object information.
2. More comprehensive evaluations with a wider range of wrist-mounted camera poses. Questions like whether the camera needs to have a clear view of the gripper, and what the implications would be if not, need addressing.

Typo:
Line 99 the the image observation

**Quality Of The Limitations Section:**

Limitations are addressed clearly

**Questions For Rebuttal:**

Could the author offer a more logical method for aligning trajectories across different robots? No need for experimental data, a detailed description would suffice.

Could the author elaborate on the degree of variation allowed for wrist-mounted poses? How well does this method generalize across diverse viewports? Is it feasible to complete the task even if the camera doesn't have a clear view of the gripper?

**Robotics Focus:**

Sufficient demonstration on hardware

**Summary Of Paper:**

The main idea of the paper is to address the challenge of transferring manipulation skills among various robotic platforms. The authors propose a novel framework to scale vision-based robotic manipulators by reusing large datasets. Their strategy involves aligning observation and action spaces across different robots using wrist cameras and a unified, modular codebase. They also bridge the domain shift by aligning the policy's internal representations, leveraging contrastive learning. The paper contributes a significant method to improve the success rate and sample efficiency for a policy using new task data from different robots, as validated by testing on a large dataset involving multiple tasks and robots.

**Summary Of Recommendation:**

The reviewer believe that this paper is above the bar of acceptance. Overall the technical method can still be improved so a weak acceptance is my recommendation before rebuttal.

## After rebuttal
The rebuttal well address the concerns of my original question. The reviewer believes that this paper can be accepted.

---

### Author Response · Authors · 2023-08-12
**Author Response to All Reviewers: Common Concerns and New Experiments**

We thank the reviewers for the thoughtful feedback. We have provided an updated version of our paper addressing the minor concerns and typos that have been brought up under our response to every reviewer. Please take a look at the highlighted portions of the PDF. In addition, please take a look at Appendix A.9 to Appendix A.11 for our new experimental results.

We will now like to briefly address some common concerns from our reviewers.

**Variation in wrist camera viewpoints**
Reviewers expressed concerns about the range of egocentric viewpoints for which our method worked. We would like to clarify that the wrist camera angles on each of the robots in our experiments have not been specifically designed or calibrated to have a similar angle. We have run additional experiments with 5 different wrist camera viewpoints outlined below, which show that our method is able to adapt to a large range of wrist angles.

**Generalization to different robots**
Reviewers mentioned concerns about the applicability of our method to different end-effectors and other robots than single-armed manipulators. We’d like to mention that even generalization across different types of single-armed parallel-jaw manipulators is a difficult open problem. We believe that there is value in leveraging commonly-available proprioception information in this setting to solve the correspondence problem. In addition, given access to pose information, CRADLE can still align trajectories and transfer notions of task completion across robots that are not limited to one arm or a parallel-jaw gripper.

**Generalization to new tasks**
Reviewers were concerned about the generalizability of our method to new tasks. We’d like to emphasize that while we do evaluate our method on a limited set of scenes, we show that CRADLE achieves high success rates in transferring experience across robots for 1. tasks which have different visual features (e.g. Pick/Place Distractor) and 2. tasks requiring different motion and understanding of dynamics than the shared dataset (e.g. New Compartment Shelf). Additionally to address concerns about task similarity with the shared dataset, we have also evaluated CRADLE on a new Pick/Place task for placing a pen in a cup, showing that CRADLE can achieve high transfer performance for tasks which require a significantly different motion for placing than seen in the shared dataset.

**Scalability of our alignment method**
Reviewers expressed concerns about the scalability of using proprioceptive information to solve the alignment problem. Although CRADLE requires access to proprioceptive state and fixed states across trajectories, this is oftentimes a weaker assumption than assuming access to environment information contained in the state. In addition, while computing distances can incur quadratic cost, this cost is trivially parallelizable and we can lower the computational complexity by sampling only a subset of trajectories per robot.



We have also run the following new experiments:

**Variability in Egocentric Camera Viewpoint**
We have 3D-printed 2 new different wrist-camera viewpoints and have collected new Pick/Place data for 5 different camera viewpoints. CRADLE achieved >70% success rate on all viewpoints for the Pick/Place distractor task, indicating that CRADLE effectively handles significant variability in camera viewpoint. Further information can be found in Appendix A.9.

**Generalization to New Tasks with OOD Motion**
We have created and evaluated a Pick/Place task variant which requires the robot to place a pen into a cup. CRADLE achieved an average of 76% success rate on this task, showing that our method can learn to transfer experience to a diverse range of tasks with potentially OOD motion.  More details are provided in the new uploaded PDF, where we have denoted this task as Scenario 3: New Container Pick/Place.

**Variability in End-Effector Size**
We have collected 50 demonstrations of the WidowX performing the new distractor pick/place task with its smaller end-effector compared to the larger, 3D-printed one. We see that CRADLE is able to achieve an average of 87% success rate on the pick/place task variants with a new end-effector. Further descriptions of the experiment can be found in Appendix A.10.

**Partial Observability of the Egocentric Viewpoint**
In order to further evaluate the applicability of our method to the case where egocentric viewpoint is partially observable, we have rerun our experiments using both egocentric and exterior camera viewpoints. CRADLE achieved an average of a 67% success rate for pick/place tasks with this method.  Further details of the experiment can be found in Appendix A.11.

---

### Decision · Program_Chairs · 2023-08-30

**Decision:**

Accept (Poster)

**Comment:**

The paper introduces a new method for robot manipulation where a single policy can be used for different robots.

After the rebuttal, the reviewers achieve consensus on accepting the paper.
Majority concerns of the reviewers have been addressed.

According to the scores of the reviewers, I recommend to accept the paper as a poster presentation.